# ATM Kinase Dead: From Ataxia Telangiectasia Syndrome to Cancer

**DOI:** 10.3390/cancers13215498

**Published:** 2021-11-01

**Authors:** Sabrina Putti, Alessandro Giovinazzo, Matilde Merolle, Maria Laura Falchetti, Manuela Pellegrini

**Affiliations:** Institute of Biochemistry and Cell Biology, IBBC-CNR, Campus Adriano Buzzati Traverso, Via Ercole Ramarini, 32, Monterotondo Scalo, 00015 Rome, Italy; alessandro.giovinazzo@ibbc.cnr.it (A.G.); matilde.merolle@ibbc.cnr.it (M.M.); marialaura.falchetti@cnr.it (M.L.F.)

**Keywords:** *ATM*, A-T patients, ATM kinase activity, ATM-KD and cancer

## Abstract

**Simple Summary:**

Ataxia telangiectasia mutated (ATM) protein plays a pivotal role in the DNA-damage response through activation of many different molecular targets. Mutations of the related gene cause Ataxia Telangiectasia (A-T) disease, characterized by neurodegeneration, immunodeficiency, and predisposition to lymphoid tumors, and they have been also found associated with several malignancies. The clinical heterogeneity of A-T can be attributed to different types of mutations that impair the expression of the protein or have a different impact on its function. In particular, extremely rare mutations that preserve the protein expression but abrogate the activity have been reported to be more dangerous in A-T patients and in mouse models. In cancer patients, these mutations have been correlated both to lymphoid and non-lymphoid tumors. The review summarizes the current knowledge on the so called “Kinase Dead” (KD) mutations of ATM protein that can be used in personalized treatments of A-T or oncologic patients.

**Abstract:**

ATM is one of the principal players of the DNA damage response. This protein exerts its role in DNA repair during cell cycle replication, oxidative stress, and DNA damage from endogenous events or exogenous agents. When is activated, ATM phosphorylates multiple substrates that participate in DNA repair, through its phosphoinositide 3-kinase like domain at the 3′end of the protein. The absence of ATM is the cause of a rare autosomal recessive disorder called Ataxia Telangiectasia characterized by cerebellar degeneration, telangiectasia, immunodeficiency, cancer susceptibility, and radiation sensitivity. There is a correlation between the severity of the phenotype and the mutations, depending on the residual activity of the protein. The analysis of patient mutations and mouse models revealed that the presence of inactive ATM, named ATM kinase-dead, is more cancer prone and lethal than its absence. *ATM* mutations fall into the whole gene sequence, and it is very difficult to predict the resulting effects, except for some frequent mutations. In this regard, is necessary to characterize the mutated protein to assess if it is stable and maintains some residual kinase activity. Moreover, the whole-genome sequencing of cancer patients with somatic or germline mutations has highlighted a high percentage of *ATM* mutations in the phosphoinositide 3-kinase domain, mostly in cancer cells resistant to classical therapy. The relevant differences between the complete absence of ATM and the presence of the inactive form in in vitro and in vivo models need to be explored in more detail to predict cancer predisposition of A-T patients and to discover new therapies for ATM-associated cancer cells. In this review, we summarize the multiple discoveries from humans and mouse models on ATM mutations, focusing into the inactive versus null ATM.

## 1. Introduction 

Ataxia telangiectasia mutated (ATM) protein is one of the three members, together with ATR and DNA-PK, belonging to the family of phosphoinositide 3-kinase (PI3K)-related kinases (PIKKs) [1] with principal roles in activating the DNA damage response (DDR). The *ATM* gene was mapped to chromosome 11q22.3 in humans [2] and to chromosome 9 in mice [3]. The *ATM* gene spanning 150 kb of genomic DNA consists of 66 exons that encode a ubiquitously expressed transcript of approximately 13 kb, which results in a 350 kDa protein of 3056 amino acids [4]. When the *ATM* gene was identified [5], it was found that the C-terminus of the predicted ATM protein contained a PI3K-like kinase domain and its signaling was driven by protein phosphorylation with a preference on serine or threonine residues followed by a glutamine (S/T-Q) [6,7]. Various S/T-Q motifs are present also in the ATM protein itself. In human cells, when DNA double strand brakes (DSBs) occur, ATM is auto-phosphorylated at multiple serine sites, and S1981 was proposed to promote the transition of the kinase inactive dimers into active monomers that orchestrate the DDR [8]. However, this process is still unclear because some studies demonstrated that S1981, and other candidates for autophosphorylation process, do not affect ATM activity in vitro [9,10] and they were considered dispensable in mouse models [11,12] (for more details see the following reviews [13,14]). Although ATM protein has a major role on DDR after DSBs in the nuclear compartment, ATM is also involved in autophagy, telomere processing, and metabolic regulation; indeed, ATM-dependent activation of cytoplasmic pathways, after reactive oxygen species (ROS), independently from DDR, has been reported [15,16,17]. The activation of ATM leads to a phosphorylation cascade of some important substrates that regulate survival and cell cycle checkpoints, such as p53 (S15), CHK2 (T68), and MDM2 (S395) [18]. Furthermore, more than 700 proteins have been predicted to be phosphorylated by ATM, in response to genotoxic agents, irradiation, or physiological DSBs [19]. Among them there are sensor, mediator, and effector proteins, which in concert cooperate to repair endogenous and exogenous DNA breakage. In line with these findings, mutations in the *ATM* gene cause an autosomal recessive rare disorder (1-9/100.000) called Ataxia Telangiectasia (A-T), first described by Syllaba and Henner [20]. A-T patients show neurodegeneration and immune dysfunction besides sensitivity to ionizing radiation, checkpoint alterations in the cell cycle, sterility, increased chromosomal breakage, telomere end fusions, and cancer predisposition [21,22]. There is a direct correlation between the severity of the phenotype and the type of mutations founded in A-T patients depending on the presence and the amount of functional ATM protein [23,24]. Indeed, classical, severe, and childhood-onset A-T is characterized by the absence of the protein due to nonsense or truncating mutations that account for 85% of mutations reported in A-T patients [25,26,27] or the presence of the protein but without kinase activity, mainly due to missense mutations [28,29]. Conversely, A-T patients with mild phenotype and adult-onset are characterized by low residual kinase activity (between 1% and 17% of functional ATM protein), due to missense or splice site mutations, and show mild neurodegeneration, but high cancer predisposition, even though delayed compared to classical A-T [28,29,30]. Mutations associated with classical A-T have not been found in hot spot regions, but they are spread in all the coding region of the gene. They were predicted to alter the conformation and the stability of the protein, mostly causing the lack of the protein and consequentially its kinase activity. More recently, *ATM* germinal and somatic mutations were also found in multiple types of cancer, such as leukemia, lymphoid malignancies, or solid tumors that show a chemotherapy resistance and adverse prognosis [31]. More than 72% of mutations of human cancer-associated *ATM* mutations are missense mutations that are highly enriched in the kinase domain of *ATM* [32]. These data highlight the crucial role of ATM in the control of various cellular processes and that not only the loss but also the presence of the protein without kinase activity contributes to the development of cancer. The purpose of this review is to summarize the information on *ATM-kinase dead* (*KD*) mutations coming from A-T or cancer patients and animal models that express an “inactivable” and/or “null” version of ATM.

## 2. ATM Kinase Dead in A-T Patients

A-T patients show a wide range of pathologic features such as cerebellar ataxia, oculocutaneous telangiectasia, variable immunodeficiency, radiosensitivity, proneness to cancer, and metabolic disorders. Cerebellar ataxia resulting from gradual loss of the Purkinje cells in the cerebellum is usually the first clinical sign and leads to progressive neuromotor deterioration. Because of this, A-T patients require wheelchairs as teenagers and they often die before their thirties, mainly from respiratory problems, recurrent infections, and cancer. Although these clinical and cellular features are considered common to all “classical” A-T patients, variations have been noted. Indeed, A-T patients show different phenotypes that vary from the severe early-onset classical disease to an adult-onset disorder with milder neurological impairment and fewer systemic symptoms [33,34,35,36,37,38,39].

Milder forms of the disease with later onset, slower clinical progression, and reduced or occasionally absent radiosensitivity have been described in several groups of A-T patients [21]. These cases, considered as mild phenotypes, are characterized by greater survival and better living conditions with less neurological damage. In most of these patients, ATM protein is expressed but with low kinase activity, due to the presence of missense mutations or small amino acids deletions that alter its functionality.

In A-T patients, mutations are distributed throughout all *ATM* coding sequence, making difficult to identify specific mutation sites that alter kinase activity function. Elucidation of the mutation nature, responsible for A-T, should give insight into the molecular and physiological bases of the disease [40]. Moreover, the mutation analysis at the A-T locus may help to estimate the prognosis of A-T patients and the predisposition of the heterozygotes to cancer, ischemic heart disease, and early mortality [41]. Many *ATM* mutations were already well characterized and some of those are summarized in the review of Taylor and colleagues [42].

Analysis of 48 A-T patients from different populations (American, Polish, Turkish, Italian, Irish, and Australian) with a “protein truncation test” suggested that PI3K domain is indispensable, and this is true also for patients with compound heterozygous mutations [43]. Another heterogeneous population of 55 A-T families with classical phenotype was analyzed by RT-PCR followed by restriction endonuclease fingerprinting [44]. In these patients were found 44 mutations with prevalence of deletions and insertions. The ATM transcript was analyzed for mutations in fibroblast and lymphoblast cell lines derived from A-T patients coming from 13 different countries. One single aa deletion (8578del3), which leads to the loss of a serine residue at the 2860 position, was found in the conserved motif within the PI3K domain typical of the protein family to which ATM belongs. A single missense mutation that leads to a Q2904G (8711A→G) substitution was also observed within the PI3K domain. The patient homozygous for this mutation shows the typical clinical A-T phenotype with neurological disturbances.

However, the presence or absence of the protein has not been characterized for all known mutations. In most of these studies, analyses have been carried out to identify only the type of mutation in the gene sequence without correlating it to the presence or absence of the protein and its kinase activity [45].

Angèle and collaborators characterized a lymphoblastoid cell line (LCL) derived from a child with a classical A-T phenotype carrying two homozygous missense *ATM* mutations 378T→A and 9022C→T [46]. They observed 50% of the normal protein amount but without kinase activity. The 378T→A is a polymorphic variant whereas 9022C→T, located in PI3K domain, leads to a missense mutation (R3008C). The functionality of the mutant protein was determined by analyzing the kinase activity, the response to ionizing radiations, and the cell cycle progression. The 9022C→T mutation in homozygosity was considered the main cause of the disease.

Reiman and colleagues established the relationship between the absence or presence of ATM kinase activity and cancer risk, based on a large cohort of 296 A-T patients from the UK and the Netherlands [29]. They observed a protective effect of ATM residual activity on patient survival and revealed that the development of childhood tumors (mainly lymphoid) in under 16-year-old A-T patients is associated almost exclusively with absence of ATM protein expression, whereas a slight delay in tumorigenesis was observed when the protein is expressed but without kinase activity. However, these last patients showed a wider range of tumor types other than of lymphoid origin.

Verhagen’s group did an extensive study on a population of 51 A-T patients, belonging to 38 families in the Netherlands, correlating the genotype to the phenotype of the patients and the respective kinase activity of the protein whether present or not [28]. They divided patients into three groups: patients without any ATM protein, patients with ATM protein but without residual ATM kinase activity, and patients with ATM protein with residual ATM kinase activity. The results of this work showed that patients with milder phenotype are those in whom the protein is present with a residual kinase activity. These patients showed a mild adult-onset of the disease, and they were cancer-prone. Patients with ATM expression but without kinase activity showed phenotypes like classical A-T in which ATM is completely absent, with a minor improvement in survival and malignancy latency until their thirties; after that period no survival or tumor free patients were reported. Deficiency of IgG_2_ subclass was found in all *ATM* null patients and in only 50% of ATM-KD patients, who also required less treatment with immunoglobulin substitution. *ATM* missense mutations resulting in mutant ATM with a defective kinase activity were distributed across the whole *ATM* coding sequence.

Recently, the mutation c3576G→A was reported to give a milder A-T phenotype despite no clear kinase activity being detected after DNA damage [47]. To characterize A-T mutations and correlate them with protein expression and function, Barone and colleagues [48] investigated the expression levels and activity of 32 *ATM* sequence variants including three in-frame deletions and a small number of known polymorphisms as positive control arising from all the missense mutations identified in UK A-T patients. They expressed ATM mutated protein in ATM null LCL obtained from A-T patients. The mutations were classified into three groups, including wild-type level of ATM kinase activity (group 1), no detectable kinase activity (group 2), or a reduced level of ATM kinase activity toward downstream targets (group 3). They observed that 10 missense mutations, resulting in protein expression without kinase activity, resided in the C-terminal of the protein, which includes the PI3K domain, and in contrast 10 missense mutations that resulted in a reduced level of ATM kinase activity were scattered across the coding region, of which six were between the N-terminal and the amino acid 1966 and four were in the C-terminal. However, in the same group were also included infrequent missense mutations that lie outside the C-terminal kinase domain.

All characterized or predicted *ATM-KD* mutations found in A-T patients (Table 1 and Figure 1) or modeled in A-T cell lines (Table 2) are reported below.

## 3. Lessons from Mouse Models: *ATM* Knockout and *ATM* Kinase Dead

Based on the discovery of the human A-T patient mutations, some animal models have been generated to recapitulate the human phenotype of the A-T disease. Considering what emerged from the human phenotype-genotype analysis, the absence of ATM protein or the presence of the kinase-dead version of ATM should lead to a murine phenotype comparable with the severe human phenotype. Indeed, the ATM null mouse models [54,55,56] showed comparable phenotypes to the human patients, except for neurodegeneration, because the cerebella of mice can largely tolerate ATM loss and maintain neuromotor functions. However, neuroinflammation, branching alteration, and displacement of Purkinje cells in the cerebellum and neurons in the brain was observed in ATM knockout mice [57,58,59,60,61,62,63]. Differently than what expected, transgenic mice expressing ATM-KD turned out to be embryonic lethal [4,64,65], whereas heterozygotes carrying an *ATM*-KD allele resembled wild-type animals, ruling out dominant-negative effects. Yamamoto and colleagues observed cytogenetic aberrations in ATM-KD embryonic stem (ES) cells with increases in chromatid breaks resulting from DSBs in G2 and M phases of the cell cycle when homologous recombination is the most active. This effect was evident in cells treated with ATM kinase inhibitors that is not observed in ATM-null cells, suggesting a role for ATM protein in the absence of kinase activity [66]. Other mouse models with the *ATM-KD* conditional allele have been generated, expressing the mutated protein in murine B cells or hematopoietic stem cells (HSCs) [32,65]. They observed a significant decrease in the frequency and number of cells in developing and mature B cell subsets in bone marrow and the spleen. Moreover, mutant murine B cells displayed more severe defects in genome stability and the increase in chromatid breaks suggested that mutant cells may have more severely impaired homologous recombination (HR). These results were confirmed by the increase of PARP inhibitor sensitivity of conditional ATM-KD cells compared with ATM-deficient cells. In contrast, defects on G1/S phase were slightly decreased in ATM-KD mice, suggesting a structural role of ATM in addition to the kinase activity function, such as the interaction with other proteins or the tethering of DNA ends, during VDJ recombination.

More studies are necessary to clarify the potential negative effect of expressing the mutated protein rather its absence. The most frequent mutations in A-T patients lead to protein truncation and instability or to protein with poor kinase activity, and very few patients have been reported to express an ATM-KD protein, suggesting a negative selective pressure on fixing this type of mutation. Conditional expression of ATM-KD in the murine nervous system showed a more pronounced neurological phenotype than ATM loss with accumulation of DSBs in cultured nervous cells and in Purkinje cells in the cerebellum [67]. Tumorigenesis was studied in mice completely lacking ATM or carrying an inactive form of the protein. Mice with *ATM* D2880A/N2885K mutations corresponding to D2870A/N2875K in humans developed blood and lymphoid cancers faster than mice with complete absence of ATM [32]. The authors suggested that ATM-KD is more oncogenic compared to the model with complete loss of ATM, and that cells are selectively hypersensitive to Topoisomerase I inhibitor. They revealed that inactive ATM prevents replication-dependent removal of Topoisomerase I-DNA adducts at the step of strand cleavage, leading to severe genomic instability and hypersensitivity to Topo Isomerase I inhibitors. This finding was important to identify ATM-KD as a marker for Topoisomerase I inhibitor-based therapy.

Recently, the same group generated two mouse models, that we will describe in the next paragraph, carrying R3016H and R3057X mutant ATM proteins that do not respond to irradiation DNA damage, but retain residual kinase activity [68,69].

All characterized kinase dead mutations found in A-T mouse model are reported below (Table 3).

## 4. ATM Kinase Dead in Cancer Patients

Cancer is a genetic disease. When mutations or DNA breaks occur, ATM protein is activated and cell proliferation is arrested through cell cycle checkpoint while waiting for the damage to be repaired or for apoptosis induction. For this reason, to allow the tumor to grow, in many cancer cells the *ATM* gene is mutated by missense mutations, especially in the PI3K domain. Those mutations mainly abrogate its function, preventing ATM from working properly through the phosphorylation of a number of proteins responsible for the DNA damage repair or programmed cell death when the lesions are too extended. We know that germline mutations, mostly deletions, that lead to the complete absence of ATM protein in A-T syndrome are associated with greatly increased risk of lymphoma and leukemia [71]. *ATM* germline heterozygous mutations were found in a big screening of 10,389 cases with genetic predisposition in 33 types of cancer, associated with loss of heterozygosity/biallelic two-hit events [72]. Six germline variants of *ATM* were coupled with somatic *ATM* mutations in prostate adenocarcinoma, rectum adenocarcinoma, stomach adenocarcinoma, esophageal carcinoma, prostate adenocarcinoma, and bladder urothelial carcinoma.

Moreover, somatic mutations spreading in the *ATM* gene have been identified in a range of cancer types, including mantle cell lymphoma (MCL), B-cell chronic lymphocytic leukemia (B-CLL), T cell prolymphocytic leukemia (T-PLL), breast, colorectal, lung, and prostate cancers [73,74].

It is known that all A-T patients with mild phenotype and less activity of ATM are prone to the development of tumors, especially leukemia. Gatti and coworkers supposed that heterozygosity for *ATM* missense mutations may result in additional phenotypic effects beyond what might result from absence or substantial reduction of ATM, as would occur with heterozygous *ATM* truncated mutations [75]. The hypothesis is that the presence of ATM mutant protein in DDR repair complex might lead to a dominant-negative effect, as demonstrated previously [76]. Missense substitutions, selected from a breast cancer cohort and A-T patients, located in the C-terminal portion of *ATM* near or within its kinase domain, were reproduced in vitro by mutagenesis of full-length *ATM* cDNA and stably transfected into A-T and control cells. Several mutant proteins, lacking ATM kinase activity, showed a dominant negative effect like the one with the breast cancer mutation *ATM* S2592C [53]. Two other mutations, one missense 7271T→G and one exon 11 splice-site mutation IVS10–6T→G, associated with high risk of breast cancer, identified by the Australian Breast Cancer Family Study/Cancer Family Registry for Breast Cancer Studies, were characterized in heterozygous cell lines by the expression and activity analyses of ATM. The 7271T→G was determined as a KD mutation in breast cancer. The results were a dominant negative effect of the mutant form of ATM and different susceptibility to radiotherapy [77]. This approach underlines the importance of screening all missense mutations founded in cancer or A-T milder phenotype patients to predict the subtle effects of the reduced ATM activity, which is not easy to predict, and to individuate the right therapy approach.

The analysis of twenty cases of MCL tumor revealed a significantly higher number of chromosomal imbalances in typical MCL with *ATM* gene alterations than in tumors with wild-type ATM, suggesting that ATM inactivation may favor increasing chromosomal instability in these lymphomas. ATM is frequently inactivated in MCL, and *ATM* gene inactivation mainly occurs by truncating mutations and missense mutations involving the PI3K domain [78].

Full genome sequence analysis of cancer cells, especially of those resistant to chemotherapy or radiation, discovered the presence of many different *ATM* somatic and germinal mutations, especially in the PI3K conserved domain where kinase domain is located [31]. Recently, an investigation on the prognostic impact of *ATM* mutational status in metastatic colorectal cancer (mCRC) was published. *ATM* is frequently mutated in colorectal cancer and, in a group of 227 Italian patients, it was observed unexpectedly that the 15% *ATM* mutated patients showed a significantly longer median overall survival compared with *ATM* wild-type tumors [79].

A mouse model, heterozygous for the most common *ATM* mutation (7636del9) that lead to kinase inactivation in A-T-patients [50,80], showed a lower incidence of thymic lymphoma but an increased susceptibility to develop tumors, including B-cell lymphomas, sarcomas, and carcinomas that have appeared in the older mice [70,81] (Table 1 and Table 3). This mutation produces near full-length detectable ATM protein that lacks protein kinase activity and was also found in T-cell prolymphocytic leukemia (T-PLL) patients. Many human cancers, reported on The Cancer Genome Atlas (TCGA) database, were *ATM* mutated and from the analysis of these mutations, Yamamoto and colleagues revealed that 72% of those are missense mutations that are highly enriched in the kinase domain [32]. Recently, the Zha group generated two mouse models carrying the *ATM* R3016H missense mutation (corresponding to *ATM* R3008H in human) in the PI3K regulatory domain (PRD) that prevent *ATM* activation and the R3057X termination mutation (corresponding to *ATM* R3047X in human) in the FATC domain of ATM that induce ATM protein instability, resulting in ATM versions lacking substrates phosphorylation after irradiation [68,69] (Table 3).

The *ATM* R3008H mutation was reported as cancer associated in CLL, MCL, diffuse large B-cell lymphoma (DLBC), and was found as the dominant negative mutation during the analysis of 140 CLL patient peripheral blood mononuclear cells (PBMCs) [74,78,82]. In contrast to the early embryonic lethality of ATMKD/KD mice, ATMR3016H (ATMR/R) mice were viable, immunodeficient, and displayed spontaneous craniofacial abnormalities and delayed lymphomagenesis compared to *ATM* knockout mice. They suggested that this mutation alters the substrates specificity and reduces the useless bond of the ATM-KD protein with its substrate preventing stalling [69]. In the second model, T and B cell defects and sterility were observed as in *ATM* knockout models, however DNA damage induced checkpoints were activated, possibly explaining the lymphomagenesis delay.

Altogether these studies demonstrated that expression of ATM-KD mice have different oncogenic features compared to ATM null mice [32,70]. Moreover, these results indicate that different approaches could be used in cancer therapy depending on the type of mutation if it abrogates ATM activity or if it prevents ATM expression or stability of the protein [73].

The multiple substrates that interact with ATM protein are the reason why missense mutation causes genomic instability and sometimes resistance to classical radiotherapy. For this reason, some work has demonstrated the hypersensitivity of ATM-KD cells during treatment with topoisomerase-1-DNA cleavage complexes (TOP1), poly-ADP-ribose polymerase (PARP), and ATR inhibitors [32,73,83,84].

All reported KD mutations found in cancer patients are reported below (Table 4).

## 5. Conclusions

ATM is a big protein that plays a central role in cell cycle control and in DDR repair by interacting with a high number of proteins. After its activation, more than 700 proteins were phosphorylated to repair DNA damage. If the damage could not be repaired, cells undergo a programmed cell death through the activation of proapoptotic proteins.

*ATM* mutations, causing the autosomal recessive hereditary disease A-T, are mainly responsible for the complete absence of the protein. The classical A-T severe phenotype is characterized by neurodegeneration, cerebellar ataxia, immunodeficiency, hypogammaglobulinemia, gonadal dysgenesis, radiosensitivity, proneness to cancer, and short life span. Some germinal mutations cause the reduction of ATM expression or activation causing mild phenotype characterized by slower progression of neurological features, longer survival, chromosomal instability, and cancer predisposition. The long-term effects of some germinal mutations in mild A-T needs to be better understood to predict the health consequences for A-T patients and possible treatments of the pathology.

Germinal mutations in *ATM* kinase domains or those that affect ATM kinase activity are extremely rare in A-T patients, probably because as suggested by mouse models, they are embryo-lethal. Evidence suggests differences in tumorigenesis onset and progress in types of tumors, and in immune system defects between patients with complete loss of ATM and those with the impairment of its sole function need further attention, and a larger number to be investigated.

All these observations mean that more studies are necessary to establish specific treatments and new gene therapies to defeat *ATM*-mutated cancer cells. The generation of inducible/conditional mouse ATM-KD models could help to study the effect and the molecular mechanism driven by inactive ATM to set up preclinical studies.

## Figures and Tables

**Figure 1 cancers-13-05498-f001:**
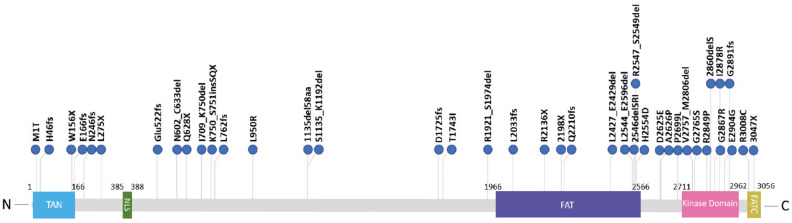
Distribution of ATM amino acid substitutions in ATM-KD patients. Schematic representation of the ATM protein 189 indicating the location of kinase dead mutations that are spread across the whole protein. The TAN (Tel1/ATM N-termi-190 nal), NLS (Nuclear Localization Signal), FAT (FRAP-ATM-TRRAP), Kinase, and FATC (FRAP-TM-TRRAP-C-terminal) domains are indicated.

**Table 1 cancers-13-05498-t001:** *ATM* mutations identified in A-T patients where ATM protein without kinase activity is expressed.

Patients	Mutations	Amino Acids	%Protein Expression Relative to Normal Level	Associated Tumors	References
A, B (hom)	7875T→G; 7876G→C	D2625E-A2626P	ND		[28,49] (15I,15II)
Aa034 (het)	8546G→C	R2849P	0%		[26]
Aa027 (het)	8599G→C	G2867R	2%		
AT41RM (hom)	8711A→G	E2904G	ND		[44] *
AT9RM (hom)	9139C→T	3047X	16.8%		[23] *
AT31RM, A-T2RM, AT35RM (het)	3403del174; 3576G→A	1135del58aa	4.8%, 5.4%, 1.0%		[23] *
AT53RM	6572ins7	2198X	11.7%		[23] *
39-4 LCL	7636del9	2546delSRI	ND **	T-ALL, TCL	[33,50]
AT173 (het)	9022C→T	R3008C	50%		[46]
(het)	8189A→C	Q2730P	≈100%		[51]
(het)	9022C→T; IVS10-6T→G	R3008C	ND	Myeloma	[52]
9 (het)	7875T→G; 7876G→C	D2625E-A2626P	ND		[28]
10, 11. I, 11.II (hom)	3576G→A	S1135_K1192del		Dermatofibrosarcoma	
12I, 12II ND	5762-2 A→T	R1921_S1974del			
13 (het)	6629delA; 8578_8580delTCT	Q2210fs 2860delS			
14.I, 14.II, 15.I, 15.II (hom)	7875T→G; 7876G→C	D2625E-A2626P		T-ALL, lymphoma	
16 (het)	7875T→G; 7876G→C; 8578_8580delTCT	D2625E-A2626P; S2860del		Breast cancer	
17.I, 17.II ND	8633T→G	I2878R		Hodgkin	
AT73	1563_1564delAG; ND	Glu522fs	>50%	B cell lymphoma C859	[29]
AT76	7660C→G; 824delT	H2554D; L275X		B-cell lymphoma	
AT82	5172dupA; ND	D1725fs		Hodgkin’s lymphoma T-ALL C813, C910, C845	
AT95-1	2250G→A; 8786+1G→A	I709_K750del; G2891fs		Lymphoma C835, C857	
AT130	468G→A; ND	W156X		Ganglioglioma	
AT134	1898+2T→G; Large Genomic Deletion	N602_C633del		Leukaemia/lymphoma C910	
AT37	6198+1G→A; ND	L2033fs		Burkitt lymphoma C837	
AT123	1882C→T; 8418+5_8delGTGA	Q628X; V2757_M2806del		Burkitt like lymphoma C833	
AT72-2	2284_2285delCT; 7280_7288del9	L762fs; L2427_E2429del		T-PLL C910	
AT21	497-ND 662+ND del; 7638_7646del9	E166fs; R2547_S2549del		Thyroid tumour C739	
AT103	2T→C; 3760ins2	M1T		lymphoma	
AT5-2 and AT5-1	2T→C; 6405_6406insTT	M1T; R2136X		Pancreatic C259; T-PLL	
AT19	5228C→T;2251-10T→G	T1743I K750_S751insSQX		T-PLL C913	
AT22-3	7638_7646del9; ND	R2547_S2549del		T-ALL	
AT8	8418+5_8delGTGA; 2849T→G	V2757_M2806del; L950R		T-ALL C910	
AT39-1 and AT39-2	138_141delTTCA; 7638_7646del9	H46fs; R2547_S2549del		T-cell lymphoma C832; C850	
ATNe45-2 (hom)	3576G→A	S1135_K1192del	>50%	Dermatofibrosarcoma protuberans	
ATNe42-1 And ATNe42-2	633T →G; ND	I2878R		M.Hodgkin	
ATNe31	738_739delinsA; 7875_7876delinsGC	N246fs; A2626P		B cell lymphoma	
ATNe10-2 And ATNe34-2 (hom)	7875_7876delinsGC	A2626P		ALL; Lymphoma	
ATNe33	7875_7876delinsGC; 8578_8580delTCT	A2626P; S2860del		Breast	
AT47	7630-ND 7788+NDdel; 8293G→A	L2544_E2596del; G2765S	≈80%	Breast C509	
AT111	2249A→G; 8293G→A	I709_K750del; G2765S		Myeloid leukaemia	
AT153	8096C→T; 2250G→A	P2699L; I709_K750del	≈50%	Lymphoma C835	

* Predicted not determined; ** in LCL from a patient with T cell tumor.

**Table 2 cancers-13-05498-t002:** *ATM* kinase dead mutations identified in A-T patients and modeled in A-T cell lines.

Mutations	Amino Acids	References
8546G→C	R2849P	[53]
8599G→C	G2867R	
7987del GTT	V2662del	
7636del9	2546delSRI	
7987delGTT	V2662del	
6056A→G	Y2019C	[48]
7181C→T	S2394L	
7660C→G	H2554D	
8189A→C	Q2730P	
8565_8566 delTGinsAA	SV2855RI	
7278_7283del6	2426delLR	
7638_7646del9	2546delSRI	
7013T→C	L2338P	
7355T→C	L2452P	
8096C→T	P2699L	
8293G→A	G2765S	
9022C>T	R3008C	
8264_8268del5	2717delGL	
9139C→T	R3047X	

**Table 3 cancers-13-05498-t003:** ATM kinase dead amino acid substitution modeled in mice.

Murine Amino Acid Substitution (Human Amino Acid Substitution)	References
D2880A/N2885K (D2870A/N2875K)	[32,64,67]
Q2740P (Q2730P)	[65]
D2899A (D2889A)	[65]
2556delSRI (2546delSRI)	[70]
R3016 (R3008H)	[69] *
R3057X (R3047X)	[68] *

* Considered with residual kinase activity in the original paper.

**Table 4 cancers-13-05498-t004:** List of common *ATM* kinase dead mutations identified in oncologic patients where ATM protein without kinase activity is expressed.

Kinase Dead Mutations in Oncologic Patients	Amino Acid	Tumors	References
7271T→G	V2424G	Breast B-CLL	[74,77]
7775C→G	S2592C	Breast	[53]
7636del9 9022C→T 9023G→A * 9139C→T * 8084G→C * 8266A→T * 8174A→T	2546delSRI R3008C R3008H R3047X G2695A K2756X D2725V	T-PLL	[74]
7181C→T	S2394L	Myeloma	[52]

* Kinase activity not determined.

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
