# Peer review of "ATM Kinase Dead: From Ataxia Telangiectasia Syndrome to Cancer"

_cancers, 2021, doi:10.3390/cancers13215498_

Round 1
Reviewer 1 Report
Please explain that when you write 'in vitro' you refer to study in humans while 'in vivo' refers to murine studies.
I found some points that need to be corrected:
- lane 111: some of the patients you refer to show absent kinase activity
- lanes 134-137: the western blots to establish the presence of the protein were routinely performed only after the first paper on the ATM gene product (see Brown et al., P.N.A.S. USA 1997, 94, 1840-1845)
A list of editing remarks is following:
- in all the text the genes must be written in cursive (typo: italic) but the protein in current characters:
ATM not ATM - lanes 32, 39, 42, 44, 46, 48, 67, 84, 113, 118, 139, 166, 168, 248, 254, 256 (twice), 260, 264, 268, 270, 273, 274, 277, 288, 291, 294, 296, 297, 299, 300, 301, 307, 311, 312, 313, 347, 355, 362, 373
Atm not Atm: lanes 228, 320, 328
lane 23: autosomal
lane 43: ..to the family..
lane 54: ..into active monomers..
lane 59: ..regulation; ..
lane 89: ..in the control..
lane 95: change 'pathologies' to 'pathologic features'
lanes 96-97: change 'susceptibility to lymphoid malignancies' with 'proneness to cancer' - some AT patients present with solid tumors
lane 100: ..already teenagers.. ..die before thirties.. (patients dying >30 are considered atypical, if not variant, cases)
lanes 106-107: cancel or move the first paragraph, it is incongruous here
lane 133: please put the reference
lane 138: ..characterized a lymphoblastoid cell line..
lane 151: please specify: do you mean ..a slight delay in tumorigenesis.. ?
lane 164: ..thirties;..
lane 171: Barone, not Giancarlo Barone
lane 185: delete A table summarizing
lane 203: ..expressing..
lane 208: ..that is not.. (it refers to effect)
lane 217: ..Atm deficient cells...
lanes 237-239: insert the bibliographic voice
lane 245: ..ATM protein..
lae 248: ..ATM gene..
lanes 309-310: Zha, not Dr Zha
lane 316: The first mutation.. which one? please specify
lane 332: ..mutation causes genomic..
lane 350: ..radiosensitivity, proneness to cancer, and short..
lane 352: ..by slower progression of neurological features, longre survival, chromosome instability..
lane 403, 407, 41, 453, 461, 465, 492, 535, 539, 566: put the year in bold
lane 426-427, 443-445, 469-470, 492, 539, 566: please complete the reference
Table 1: please correct AT523RM in AT53RM (7th lane)
Author Response
We thank the reviewer for his/her useful suggestions.
Below our point by point response:
Please explain that when you write 'in vitro' you refer to study in humans while 'in vivo' refers to murine studies
We modified the text specifying the models.
All the suggested corrections, references and editing have been done. Major changes are written in red.
Reviewer 2 Report
This is a comprehensive review of various types of ATM mutations and their impacts on A-T and cancer predisposition. In particular, this review looks into the consequences of the kinase-dead proteins, which act sometimes worse than the absence of protein. In general, this paper is well written and I have a few comments or modest suggestions.
- I would modestly suggest to add a schematic diagram of ATM protein structure and the locations of mutations.
- In introduction, authors may add citation of following works, which I believe are the milestone works to establish that ATM is really protein kinase.
- Banin, S.; Moyal, L.; Shieh, S.; Taya, Y.; Anderson, C.W.; Chessa, L.; Smorodinsky, N.I.; Prives, C.; Reiss, Y.; Shiloh Y. et al. En-hanced phosphorylation of p53 by ATM in response to DNA damage. Science 1998, 281, 1674-1677.
- Canman, C.E.; Lim, D.S.; Cimprich, K.A.; Taya, Y.; Tamai, K.; Sakaguchi, K.; Appella, E., Kastan, M.B.; Siliciano, J.D. Activa-tion of the ATM kinase by ionizing radiation and phosphorylation of p53. Science 1998, 281, 1677-1679.
- In page 2, line 54-55: “that translocate to the nucleus and participate on DNA damage repair”. In my understanding ATM is located in the nucleus before activation, not translocating after activation. In addition, ATM participate in DNA damage response, not limited to DNA damage repair.
- In page 7, line 209: The meaning of “selective function of the kinase inhibited Atm protein” is somewhat obscure.
- In page 7, line 219: The meaning of “a plausible structural role of Atm during VDJ recombination” is somewhat obscure.
- In page 7, line 237-239: “Recently, the same group … residual kinase activity in vivo” requires citation.
- In page 8, line 260-265: This part is describing somatic mutation in cancer but unclear. Do the sentences “Moreover, somatic mutations, … cancer types” and “ATM is mutated … prostate cancers” mean the same thing? If not the former needs citation. Additionally, in the sentence between them, “ATM-mediated apoptosis” might sound better.
- In page 8, line 266-267: Does it mean “We know that all A-T patients with mild phenotype and less activity of ATM are more prone to the development of tumors especially leukemia”?
- Consistent style for the notation of amino acid or its substitution should be used. (There are S1981 as well as serine 15 and Glu2904Gly as well as R3008C.)
- Other minor corrections.
In Table 4: 3rd row may need correction.
Reference 49 needs correction.
Page 7, line 204: “indicating lack of” -> “excluding” or “ruling out”?
Page 8, line 246: “trough” -> “through”.
Author Response
We thank the reviewer for his/her useful suggestions.
Below our point by point response. Major changes are written in red.
- I would modestly suggest to add a schematic diagram of ATM protein structure and the locations of mutations.
We added figure 1 showing the location of mutations reported in A-T patients carrying ATM kinase dead and we inserted the suggested references
- In introduction, authors may add citation of following works, which I believe are the milestone works to establish that ATM is really protein kinase.
- Banin, S.; Moyal, L.; Shieh, S.; Taya, Y.; Anderson, C.W.; Chessa, L.; Smorodinsky, N.I.; Prives, C.; Reiss, Y.; Shiloh Y. et al. En-hanced phosphorylation of p53 by ATM in response to DNA damage. Science 1998, 281, 1674-1677.
- Canman, C.E.; Lim, D.S.; Cimprich, K.A.; Taya, Y.; Tamai, K.; Sakaguchi, K.; Appella, E., Kastan, M.B.; Siliciano, J.D. Activa-tion of the ATM kinase by ionizing radiation and phosphorylation of p53. Science 1998, 281, 1677-1679.
The references were added in the text
- In page 2, line 54-55: “that translocate to the nucleus and participate on DNA damage repair”. In my understanding ATM is located in the nucleus before activation, not translocating after activation. In addition, ATM participate in DNA damage response, not limited to DNA damage repair.
Thanks the sentence was misleading we removed “translocate to the nucleus” and we added two related recent reviews on ATM activation mechanisms
- In page 7, line 209: The meaning of “selective function of the kinase inhibited Atm protein” is somewhat obscure.
We changed as follow: “suggesting a role for Atm protein in the absence of kinase activity”
- In page 7, line 219: The meaning of “a plausible structural role of Atm during VDJ recombination” is somewhat obscure.
We changed as follow: “a structural role of Atm in addition to the kinase activity function, such as the interaction with other proteins or the tethering of DNA ends, during VDJ recombination.”
- In page 7, line 237-239: “Recently, the same group … residual kinase activity in vivo” requires citation.
The two references were added:
Milanovic, M., Shao, Z., Estes, V.M., Wang, X.S., Menolfi, D., Lin, X., Lee, B.J., Xu, J., Cupo, O.M., Wang, D., et al. (2021a). FATC Domain Deletion Compromises ATM Protein Stability, Blocks Lymphocyte Development, and Promotes Lymphomagenesis. J. Immunol. 206, 1228–1239.
Milanovic, M., Houghton, L.M., Menolfi, D., Lee, J.H., Yamamoto, K., Li, Y., Lee, B.J., Xu, J., Estes, V.M., Wang, D., et al. (2021b). The Cancer-Associated ATM R3008H Mutation Reveals the Link between ATM Activation and Its Exchange. Cancer Res. 81, 426–437.
- In page 8, line 260-265: This part is describing somatic mutation in cancer but unclear. Do the sentences “Moreover, somatic mutations, … cancer types” and “ATM is mutated … prostate cancers” mean the same thing? If not the former needs citation. Additionally, in the sentence between them, “ATM-mediated apoptosis” might sound better.
The two sentences were combined and the two related references were included: “Moreover, somatic mutations spreading in the ATM gene have been identified in a range of cancer types, including mantle cell lymphoma (MCL), B-cell chronic lymphocytic leukemia (B-CLL), T cell prolymphocytic leukemia (T-PLL), breast, colorectal, lung, and prostate cancers (Jette et al., 2020; Stankovic et al., 2002)”.
- In page 8, line 266-267: Does it mean “We know that all A-T patients with mild phenotype and less activity of ATM are more prone to the development of tumors especially leukemia”?
Thank you we modified according to the suggestion
- Consistent style for the notation of amino acid or its substitution should be used. (There are S1981 as well as serine 15 and Glu2904Gly as well as R3008C.)
The style was uniformed
- Other minor corrections.
The minor corrections were inserted